# Assessment of In-Hospital Pain Control after Childbirth and Its Correlation with Anxiety in the Postpartum Period: A Cross-Sectional Study at a Single Center in the USA

Clara G. Olson [1], John R. Soehl [2,3], Zachary N. Stowe [4] and Kathleen M. Antony [2,*]

1    Department of Medical Education, School of Medicine & Public Health, University of Wisconsin, 750 Highland Avenue, Madison, WI 53726, USA
2    Department of Obstetrics and Gynecology, University of Wisconsin-Madison, 202 South Park Street, Madison, WI 53706, USA
3    Division of Maternal Fetal Medicine, Department of Obstetrics and Gynecology, Warren Alpert Medical School of Brown University, 222 Richmond Street, Providence, RI 02903, USA
4    Department of Psychiatry, University of Wisconsin-Madison, 6001 Research Park Boulevard, Madison, WI 53706, USA
*    Correspondence: kantony@wisc.edu

**Abstract:** Anxiety is common during the antepartum, intrapartum, and postpartum period. While the relationship between obstetric pain and depression is well characterized, there are few publications examining the relationship between obstetric pain and anxiety. Our objective was to characterize the association, if any, between postpartum pain and anxiety. This was a survey-based cross-sectional study. The general anxiety disorder (GAD)-7 and American Pain Society patient outcome questionnaire (APS-POQ) were completed by 64 postpartum participants at hospital discharge. Associations between anxiety and pain control were assessed. Participants with moderate to severe scores (greater or equal to 10) on the GAD-7 had more maximum pain scores (0 to 10 scale) in the severe range (greater or equal to 7) in the first ($p = 0.049$) and second ($p = 0.010$) 24 h periods after delivery and were more likely to have spent more time in severe pain within these time frames ($p = 0.007$ and $p = 0.010$, respectively). Similar relationships were observed when classifying anxiety ordinally. In conclusion, higher postpartum pain scores were associated with greater anxiety in the postpartum period.

**Keywords:** anxiety; obstetrics; postpartum pain; pain expectations; perinatal anxiety; postpartum




## 1. Introduction

Anxiety is a prevalent condition that affects up to 33.7% of the population at some point during their lifetime with a median age of onset of 17 [1–3]. Anxiety during pregnancy and postpartum is common; 35% of pregnant people report anxiety during pregnancy, 17% immediately after childbirth, and 20% at six weeks postpartum [3]. Feelings of anxiety can interfere with daily activities and be difficult to control. Postpartum anxiety has been shown to be associated with lower maternal self-confidence [4] and can have adverse effects on child development through mother–infant interactions, bonding, breastfeeding, sleep, mental development, and infant temperament [5]. Postpartum anxiety negatively impacts both the mother and child; therefore, it is important to address factors that contribute to the development of anxiety in the hopes of minimizing these effects.

Although childbirth is typically a joyful time, it is often painful and stressful [3]. Labor pain has been shown to be a predictor of perceiving childbirth as traumatic [6] and may cause anxiety and exhaustion to mothers which can have adverse effects on the progress of labor [7]. Severe pain in the postpartum period can also predict persistent pain and adverse mental health outcomes; patients who had severe acute postpartum pain had a 2.5-fold increased risk of persistent pain at 8 weeks post-delivery and a 3.0-fold increased risk of postpartum depression compared to those with mild postpartum pain [8]. It has also been

found that reducing intrapartum pain with epidural analgesia led to a significant decline in anxiety and fatigue during labor [9]. These observations suggest the need to focus on treating pain during labor and in the postpartum period to prevent persistent pain and negative impacts on long-term mental health. In the last decade, research on anxiety has increased, but studies evaluating pain and obstetric anxiety remain scarce [10]. Additional attention is needed to identify the factors contributing to postpartum anxiety in order to potentially mitigate them.

Here we sought to examine the relationship between postpartum pain control and postpartum anxiety. We hypothesized that less adequate pain control in the postpartum period would be associated with higher levels of maternal anxiety in the immediate and longer-term postpartum period.

## 2. Materials and Methods

This survey-based cross-sectional study was approved by the Institutional Review Board at UnityPoint Health Meriter Hospital (IRB#2019-016), which is the clinical home of the University of Wisconsin-Madison Department of Obstetrics and Gynecology. Patients admitted to the postpartum unit at UnityPoint Health-Meriter Hospital between 24 May 2021–30 July 2021 were approached by the obstetric resident physicians, midwives, or nurse practitioners to see if they were willing to be contacted about this study. Those who were willing to be contacted were briefly screened for exclusion and inclusion criteria by review of the electronic health record (Epic CareLink 2020, Epic Systems Corporation, Verona, WI, USA). Exclusion criteria included the following: the current pregnancy resulted in stillbirth or intrauterine fetal demise, delivery via cesarean hysterectomy, lack of internet access, or chronic opioid use due to opioid use disorder or chronic pain. Inclusion criteria consisted of a minimum age of 18, able to read and understand English, access to the internet using a smartphone, computer, or other device, and willingness to consent to completion of two online surveys distributed via email. Patients who appeared eligible were then called by one researcher (CO). Those who were interested in participating were emailed a pre-screening survey with eligibility and consent questions. If the patient was not eligible based on the eligibility questions, or if they did not consent to participate, the survey ended, and they did not proceed. Those who met criteria and who consented were enrolled. Once enrolled, the general anxiety disorder (GAD)-7 and American Pain Society patient outcome questionnaire (APS-POQ) survey and other questions about pain were sent to their email via REDCap (Research Electronic Data Capture, v11.0.3, Nashville, TN, USA) with instructions to complete this on the day of discharge [11,12]. Up to three reminder emails were sent. At 6 weeks postpartum, an email with the GAD-7 survey was sent, with up to three reminder emails. Flyers were also posted in the patient rooms on the postpartum unit which had a QR code which linked to the survey and eligibility criteria form; thus, if an interested patient wanted to participate, they could access the survey independently and proceed via the same methods as mentioned previously. Notably, the original plan was for the researcher to speak directly with the patient in-person for both introducing the study and to offer to assist with completion of the survey. However, due to COVID-related restrictions, the researcher was not able to be physically present on the postpartum unit; thus, the protocol and regulatory boards were updated to the recruitment approach discussed here.

The APS-POQ is designed to assess adults' pain. It queries one's maximum pain in the prior 24 h and to what degree pain interferes with one's activities, sleep, and mood [13,14]. This questionnaire has high reliability and validity [13–15]. It also queries pain relief. Pain scores reported here were determined by the APS-POQ and were classified as the following: 0 = no pain, 1–3 = mild pain, 4–6 = moderate pain, and 7–10 = severe pain. The degree to which pain interferes with activity was completed on a scale from 0–10 where 0 signifies that the pain does not interfere with a given activity and 10 signifies that the pain completely interferes with a given activity [13,14]. The APS-POQ also queries satisfaction with pain treatment [13,14].

The GAD-7 is frequently used as a diagnostic self-report for patients experiencing symptoms of anxiety over the preceding two weeks [16]. The GAD-7 has been validated in the general population and pregnant populations [17,18]. The GAD-7 was scored as follows: no anxiety (GAD-7 score less than 5), mild anxiety (GAD-7 score 5 to 9), moderate anxiety (GAD-7 score 10 to 14), and severe anxiety (GAD-7 score greater than 15). For analyses with anxiety as a dichotomous variable, the presence of moderate or severe anxiety was determined by a score of 10 or greater on the GAD-7.

Following enrollment, the diagnosis of pre-existing anxiety and depression were manually extracted from the medical record. A history of anxiety was defined as having a generalized anxiety disorder, anxiety, social anxiety, or situational anxiety disorder listed in their admission history and physical (H&P) note or in their "problem list" in the electronic health record. Panic disorder was also included. Diagnoses that did not explicitly include 'anxiety' (e.g., obsessive compulsive disorder (OCD), post-traumatic stress disorder (PTSD)) were not included. This was largely done to specifically focus on 'anxiety'. Notably, both OCD and PTSD are no longer listed under anxiety disorders in the *Diagnostic and Statistical Manual of Mental Disorders, Fifth Edition* (DSM V) DSM V [19]. The mention of any history of a depressive disorder documented in the H&P, or their "problem list" was also documented, such as "depression", "major depressive episode", or "postpartum depression". Diagnoses that did not explicitly mention "depression" (e.g., adjustment disorder) were not included. Other maternal characteristics such as age, race, insurance, education, parity, prior miscarriages or abortions, BMI, and tobacco use were extracted from the medical record.

Fetal anomalies (such as spina bifida, ambiguous genitalia, and suspected VSD) diagnosed before birth, delivery information including mode of delivery, type or presence of cesarean or labor analgesia, vaginal lacerations, intraamniotic infection, gestational age, length of stay, preterm delivery, and NICU admission information were also extracted from the medical record.

Study data were collected and managed using REDCap electronic data capture tools hosted at the University of Wisconsin-Madison. REDCap [11,12] (Research Electronic Data Capture) is a secure, web-based software platform designed to support data capture for research studies. It also distributes surveys with calendar-enabled features and stores survey data.

*Statistical Analysis*

Maternal and obstetric characteristics for parturients with a GAD-7 score < 10 and ≥10 were compared using Student's *t*-test, $X^2$, or Fisher's exact test, as appropriate. Associations between GAD-7 scores and pain measures were similarly calculated using $X^2$, Fisher's exact test, or the Wilcoxon–Rank sum for non-normally distributed continuous variables. Our sample size precluded adjustment for confounding variables. An a priori power calculation suggested that a minimum sample size of 400 would be required, assuming 10% of participants had severe pain and that 30% of those with severe pain would have moderate or severe anxiety compared to 10% of those without severe pain. However, due to COVID-related restrictions on access, both recruitment time and access level were abbreviated and the sample size here is less. A post hoc power analysis found that a sample size from 55 to 65 provided 80% power with an alpha of 95%. All statistical analyses were completed using STATA 16.0 (StataCorp. 2019. Stata Statistical Software: Release 16. College Station, TX, USA, StataCorp LLC). Data is available upon reasonable request to the corresponding author.

## 3. Results

Of the 125 who gave permission to be contacted, 106 met the criteria for inclusion and 101 consented to complete the survey and have their record reviewed for demographic and obstetrical variables. Of these, 64 participants completed discharge surveys (Figure 1). No

participants self-enrolled via the QR code; all were enrolled by obtaining permission from the rounding provider.

## Participant Enrollment

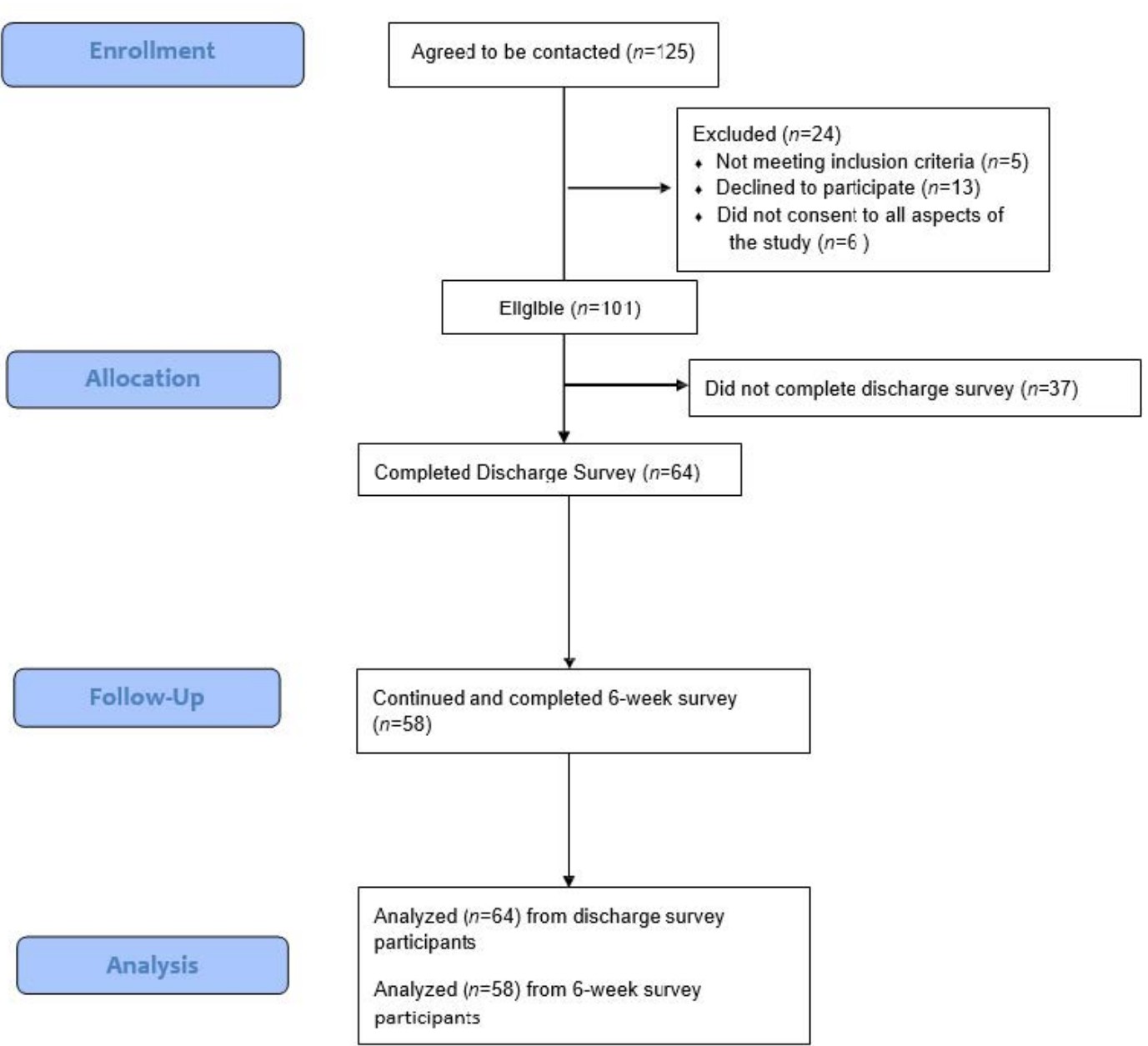

**Figure 1.** Participant enrollment.

Maternal characteristics of parturients with GAD-7 scores at hospital discharge <10 and ≥10 were similar except for insurance provider; more patients with anxiety had Tricare, which covers uniformed service members, retirees, and their families ($p < 0.001$). Notably, the presence of mood disorders, such as anxiety, depression, or both, were not statistically different between parturients with GAD-7 scores at discharge of <10 versus ≥10 (Table 1). Delivery characteristics were also similar except more patients with anxiety experienced intraamniotic infection ($p = 0.015$) (Table 2).

**Table 1.** Maternal characteristics.

| Characteristic | | GAD-7 < 10 (*n* = 57) | GAD-7 ≥ 10 (*n* = 7) | *p*-Value * |
|---|---|---|---|---|
| Age, mean (SD) | | 32.4 (3.3) | 33.9 (4.1) | 0.281 |
| Race/Ethnicity, *n* (%) | Non-Hispanic White | 53 (93.0) | 5 (71.4) | 0.065 |
| | Non-Hispanic Black or African American | 0 (0) | 0 (0) | |
| | Asian American | 4 (7.0) | 2 (28.6) | |
| | Latinx | 0 (0) | 0 (0) | |
| | Other/unknown | 0 (0) | 0 (0) | |
| Married or partnered, *n* (%) | | 54 (94.7) | 7 (100.0) | 0.534 |
| Insurance, *n* (%) | Private insurance | 56 (98.2) | 5 (71.4) | <0.001 |
| | Medicaid | 1 (1.8) | 0 (0) | |
| | Tricare | 0 (0) | 2 (28.6) | |
| Education, *n* (%) | Less than high school | 0 (0) | 0 (0) | 0.810 |
| | High school education | 1 (1.75) | 0 (0) | |
| | Some college | 3 (5.26) | 0 (0) | |
| | College degree | 3 (5.26) | 0 (0) | |
| | Masters degree or higher | 50 (87.7) | 7 (100) | |
| Parity, *n* (%) | Primiparous | 31 (54.4) | 4 (57.1) | 0.890 |
| | Multiparous | 26 (45.6) | 3 (42.9) | |
| Prior miscarriage or abortion, *n* (%) | | 12 (21.1) | 2 (28.6) | 0.650 |
| Pre-pregnancy maternal BMI [†], mean ± SD | | 30.1 (6.2) | 33.2 (8.2) | 0.227 |
| Tobacco use, *n* (%) | | 0 (0) | 0 (0) | NA |
| Mood disorders, *n* (%) | Anxiety | 8 (14.0) | 1 (14.3) | 0.730 |
| | Depression | 3 (5.3) | 0 (0) | |
| | Anxiety and depression | 8 (14.0) | 2 (28.6) | |
| | Neither | 38 (66.7) | 4 (57.1) | |

* Chi-square test used for categorical variables and Student's *t*-test for continuous variables. [†] Body mass index.

Parturients with moderate to severe scores (greater or equal to 10) on the GAD-7 had more maximum pain scores (0 to 10 scale) in the severe range (greater or equal to 7) in the first (*p* = 0.049) and second (*p* = 0.010) 24 h periods after delivery. They were more likely to have spent a higher percentage of their time in severe pain within these time frames (*p* = 0.007, and *p* = 0.010, respectively) (Table 3). They also had more difficulty doing activities in bed (*p* = 0.009), doing activities out of bed (*p* = 0.007), falling asleep (*p* < 0.001), and staying asleep (*p* < 0.002) (Figure 2). Information they were given about their treatment plan was perceived as less helpful (*p* < 0.001), they had less adequate pain control after giving birth (*p* = 0.003), and overall were less satisfied with their pain treatment (*p* = 0.003) (Figure 3).

When classifying anxiety as no anxiety (GAD-7 score less than 5), mild anxiety (GAD-7 score 5 to 9), moderate anxiety (GAD-7 score 10 to 14), and severe anxiety (GAD-7 score greater than 15), similar results were observed. (Table 4.) Specifically, increasing anxiety severity was inversely associated with the lowest pain score being less than or equal to 3 (*p* = 0.002) and directly associated with any severe pain in the second 24 h time period after cesarean (*p* = 0.047).

**Table 2.** Delivery Characteristics.

| Delivery Characteristics | | GAD-7 < 10 (*n* = 57) | GAD-7 ≥ 10 (*n* = 7) | *p*-Value * |
|---|---|---|---|---|
| Mode of delivery, *n* (%) | Vaginal | 34 (59.7) | 1 (14.3) | 0.053 |
| | Vacuum | 1 (1.8) | 1 (14.3) | |
| | Forceps | 0 (0) | 0 (0) | |
| | Scheduled cesarean | 9 (15.8) | 2 (28.6) | |
| | Urgent cesarean | 12 (21.1) | 2 (28.6) | |
| | Emergency cesarean | 1 (1.8) | 1 (14.3) | |
| Cesarean birth analgesia, ‡ *n* (%) | Epidural | 9 (40.9) | 1 (20) | 0.552 |
| | Spinal | 11 (50.0) | 3 (60.0) | |
| | Combined spinal/epidural | 1 (4.6) | 1 (20.0) | |
| | General anesthesia | 1 (4.6) | 0 (0) | |
| Labor analgesia, *n* (%) | No pain medications | 3 (6.4) | 0 (0) | 0.055 |
| | IV pain medications only | 2 (4.3) | 1 (25.0) | |
| | Epidural only | 28 (49.6) | 0 (0) | |
| | IV pain meds and epidural | 14 (29.8) | 3 (75.0) | |
| Vaginal lacerations, ‡ *n* (%) | No laceration | 4 (11.4) | 0 (0) | 0.152 |
| | 1st degree | 8 (22.9) | 0 (0) | |
| | 2nd degree | 21 (60.0) | 1 (50.0) | |
| | 3rd degree | 2 (5.7) | 1 (50.0) | |
| | 4th degree | 0 (0) | 0 (0) | |
| Intraamniotic infection, *n* (%) | | 1 (1.1) | 1 (14.3) | **0.015** |
| Gestational age (weeks), mean (SD) | | 39.0 (1.4) | 39.4 (0.8) | 0.399 |
| Length of stay > 72 h, *n* (%) | | 15 (26.32) | 5 (57.14) | 0.092 |
| Preterm delivery, *n* (%) | | 6 (10.5) | 0 (0) | 0.367 |
| Fetal anomalies present, *n* (%) | | 5 (8.8) | 0 (0) | 0.414 |
| NICU § Admission, *n* (%) | | 3 (5.4) | 0 (0) | 0.530 |

* Chi-square test used for categorical variables and Student's *t*-test for continuous variables. ‡ Denominator is only cesarean births or for labor, where appropriate. § Neonatal intensive care unit.

**Table 3.** Associations between pain and anxiety. Minimum, maximum pain scores in the first and second 24 h time period and for the whole hospitalization, and percentage of time spent in severe pain, per patient survey.

| Variable | | GAD-7 < 10 (*n* = 57) | GAD-7 ≥ 10 (*n* = 7) | *p*-Value * |
|---|---|---|---|---|
| First 24 h post-delivery(*n* = 64) | Maximum pain score ≥ 7, *n* (%) * | 19 (33.3) | 5 (71.4) | **0.049** |
| | % of time spent in severe pain, median (IQR) † | 10 (0–10) | 10 (10–60) | **0.007** |
| | 20% of time or more spent with severe pain, *n* (%) | 12 (21.0) | 3 (42.9) | 0.199 |
| | Lowest pain score ≤ 3, *n* (%) | 51 (89.5) | 3 (42.9) | **0.001** |

**Table 3.** *Cont.*

| Variable | | GAD-7 < 10 (*n* = 57) | GAD-7 ≥ 10 (*n* = 7) | *p*-Value * |
|---|---|---|---|---|
| Second 24 h post-delivery | Maximum pain score ≥ 7, *n* (%) | 14 (24.6) | 5 (71.4) | **0.010** |
| | % of time spent in severe pain, median (IQR) | 0 (0–10) | 10 (10–50) | **0.010** |
| | 20% of time or more spent with severe pain, *n* (%) | 10 (17.5) | 2 (28.6) | 0.481 |
| | Lowest pain score ≤ 3, *n* (%) | 54 (94.7) | 5 (71.4) | **0.030** |
| Overall for hospitalization | Maximum pain score ≥ 7, *n* (%) | 24 (42.11) | 5 (71.43) | 0.141 |
| | 20% of time or more spent with severe pain, *n* (%) | 15 (26.32) | 3 (42.86) | **0.358** |
| | Lowest pain score ≤ 3, *n* (%) | 56 (98.25) | 5 (71.43) | **0.002** |
| | Total doses of opioid medications, † | 0 (0–1) | 3 (0–17) | **0.022** |
| | Perceived adequacy of pain control, median (IQR) †‡ | 9 (8–10) | 7 (2–8) | **0.003** |
| | Satisfaction with pain control, median (IQR) †§ | 10 (8–10) | 6 (5–8) | **0.003** |

* Chi-square test used for categorical variables. † Mann–Whitney U-test for ordinal variables ("percentage of time spent in severe pain" was on an ordinal scale) and for non-normally distributed data (example: total opioid doses). ‡ Measured on a 0–10 scale with 0 being completely inadequate and 10 being completely adequate. § Measured on a 0–10 scale with 0 being extremely dissatisfied and 10 being extremely satisfied.

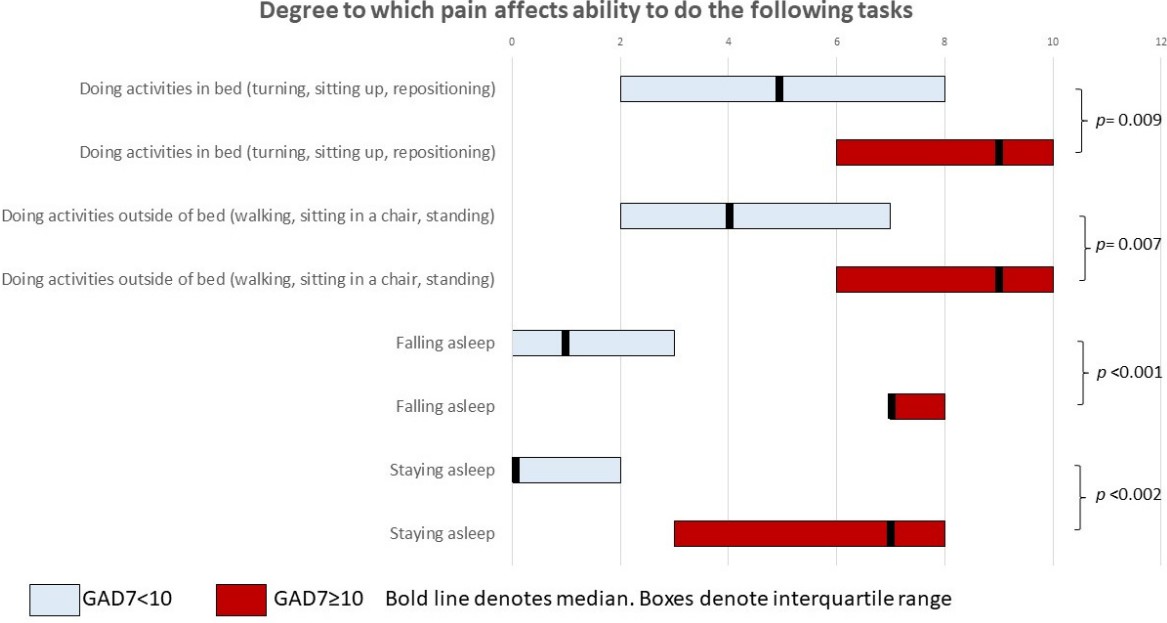

**Figure 2.** Responses to the question: "Pain can affect how we do daily tasks. Please choose the option that best describes how much your pain after giving birth interfered with or prevented you for doing the following things while in the hospital. 0 = did not interfere at all, 10 = completely interfered". Median and interquartile ranges of responses, Mann–Whitney U-test performed. GAD-7 results < 10 are shown in light blue, GAD-7 results ≥ 10 are shown in red. Bold line denotes the median and boxes denote the interquartile range.

**Pain treatment participation, information about treatment, and satisfaction with pain control**

**Figure 3.** Responses to questions about patient participation in decisions related to pain, information about pain treatment, and satisfaction with pain control. * Responses were 1 = always, 2 = most times, 3 = sometimes, 4 = rarely, 5 = never. † Responses were on scale of 0–10 (0 being not helpful at all, 10 being very helpful). ‡ Responses were on scale of 0–10 (0 = no pain, 10 = extreme pain). § Responses were on scale of 0–10 (0 = completely inadequate, 10 = completely adequate). ‖ Responses were on scale of 0–10 (0 = extremely dissatisfied, 10 = extremely satisfied). ⸕ Responses were on a scale of 0–100% in intervals of 10 (0 = no relief at all, 100% = complete relief). Median and interquartile ranges of responses, Mann–Whitney U-test performed. GAD-7 results < 10 are shown in light blue, GAD-7 results ≥ 10 are shown in red. Bold line denotes the median and boxes denote the interquartile range.

**Table 4.** Associations between pain and anxiety with anxiety categorized as no anxiety (GAD-7 < 5), mild anxiety (GAD-7 = 5–9), moderate anxiety (GAD-7 = 10–14), or severe anxiety (GAD-7 = 15–21).

| | | GAD-7 < 5 N = 44 | GAD-7 5–9 N = 13 | GAD-7 10–14 N = 5 | GAD-7 15–21 N = 2 | *p* |
|---|---|---|---|---|---|---|
| First 24 hours postpartum | Any severe pain, *n* (%) | 17 (38.6) | 2 (15.4) | 3 (60.0) | 2 (100) | 0.067 |
| | 20% of time or more spent with severe pain, *n* (%) | 10 (22.7) | 2 (15.4) | 2 (40.0) | 1 (50.0) | 0.566 |
| | Lowest pain score ≤ 3, *n*(%) | 40 (90.9) | 11 (84.6) | 3 (60.0) | 0 (0) | **0.002** |
| Second 24 hours postpartum | Any severe pain | 10 (22.7) | 4 (30.8) | 3 (60.0) | 2 (100.0) | **0.047** |
| | 20% of time or more spent with severe pain | 8 (18.2) | 2 (15.4) | 1 (20.0) | 1 (50.0) | 0.707 |
| | Lowest pain score ≤ 3 | 42 (95.4) | 12 (92.3) | 4 (80.0) | 1 (50.0) | 0.085 |

Parturients with preexisting anxiety did not have higher rates severe pain, percent of time with severe pain, or less mild pain than those without a history of anxiety, nor did they have different perceptions of inadequate pain control. There were 27 participants who described themselves as anxious, and they did have non-statistically significant less mild pain in the first 24 h ($p = 0.053$), but they did not have higher rates of severe pain or percent of time with severe pain. Parturients whose infants who had health concerns (e.g., fetal anomalies) or NICU admissions postpartum did not have higher rates of anxiety.

Of the 64 who completed the discharge survey, 58 completed the 6-week postpartum GAD-7 survey, with a response rate of 90.6%. The 6-week survey was completed on average 41 days after the discharge survey (range 40–55 days). Those with GAD-7 scores $\geq$ 10 were less satisfied with their pain control ($p = 0.049$) (Tables 5 and 6).

**Table 5.** Baseline and delivery characteristics of participants in the 6-week postpartum survey.

| Characteristic | | GAD-7 < 10 (*n* = 49) | GAD-7 $\geq$ 10 (*n* = 9) | *p*-Value * |
|---|---|---|---|---|
| Age, mean (SD) | | 32 (3.6) | 31.4 (2.8) | 0.366 |
| Race/ethnicity, *n* (%) | Non-Hispanic White | 44 (89.8) | 8 (88.9) | 0.935 |
| | Non-Hispanic Black or African American | 0 (0) | 0 (0) | |
| | Asian American | 5 (10.2) | 1 (11.1) | |
| | Latinx | 0 (0) | 0 (0) | |
| | Other/unknown | 0 (0) | 0 (0) | |
| Married or partnered, *n* (%) | | 47 (95.9) | 8 (88.9) | |
| Insurance, *n* (%) | Private insurance | 47 (95.9) | 8 (88.9) | 0.361 |
| | Medicaid | 1 (2.0) | 0 (0) | |
| | Tricare | 1 (2.0) | 1 (11.1) | |
| Education, *n* (%) | Less than high school | 0 (0) | 0 (0) | 0.691 |
| | High school education | 1 (2.0) | 0 (0) | |
| | Some college | 3 (6.10) | 0 (0) | |
| | College degree | 2 (4.1) | 1 (11.1) | |
| | Masters degree or higher | 43 (87.8) | 8 (88.9) | |
| Parity, *n* (%) | Primiparous | 27 (55.1) | 5 (55.6) | 0.980 |
| | Multiparous | 22 (44.9) | 4 (44.4) | |
| Prior miscarriage or abortion, *n* (%) | | 8 (18.4) | 2 (22.2) | 0.786 |
| Pre-pregnancy maternal BMI [†], mean $\pm$SD | | 30.7 (6.5) | 30.6 (7.6) | 0.947 |
| Tobacco use, *n* (%) | | 0 (0) | 0 (0) | NA |
| Mood disorders, *n* (%) | Anxiety | 6 (12.2) | 2 (22.2) | 0.696 |
| | Depression | 2 (4.1) | 0 (0) | |
| | Anxiety and depression | 7 (14.3) | 1 (22.2) | |
| | Neither | 34 (69.4) | 5 (55.6) | |
| Mode of delivery, *n* (%) | Vaginal | 29 (59.2) | 4 (44.4) | 0.397 |
| | Vacuum | 1 (2.0) | 1 (11.1) | |
| | Forceps | 0 (0) | 0 (0) | |
| | Scheduled cesarean | 8 (16.3) | 1 (11.1) | |
| | Urgent cesarean | 10 (20.4) | 2 (22.2) | |
| | Emergency cesarean | 1 (2.0) | 1 (11.1) | |
| Cesarean birth analgesia, [†] *n* (%) | Epidural | 7 (36.8) | 1 (25.0) | **0.015** |
| | Spinal | 11 (57.9) | 1 (25.0) | |
| | Combined spinal/epidural | 0 (0) | 2 (50.0) | |
| | General anesthesia | 1 (5.3) | 0 (0) | |

**Table 5.** *Cont.*

| Characteristic | | GAD-7 < 10 (*n* = 49) | GAD-7 ≥ 10 (*n* = 9) | *p*-Value * |
|---|---|---|---|---|
| Labor analgesia, *n* (%) | No pain medications | 3 (7.5) | 0 (0) | 0.404 |
| | IV pain medications only | 3 (7.5) | 0 (0) | |
| | Epidural only | 23 (57.5) | 3 (42.9) | |
| | IV pain meds and epidural | 11 (27.5) | 4 (57.1) | |
| Vaginal lacerations, ‡ *n* (%) | No laceration | 3 (10.0) | 1 (20.0) | 0.285 |
| | 1st degree | 8 (26.7) | 0 (0) | |
| | 2nd degree | 18 (60.0) | 3 (60.0) | |
| | 3rd degree | 1 (3.3) | 1 (20.0) | |
| | 4th degree | 0 (0) | 0 (0) | |
| Intraamniotic infection, *n* (%) | | 2 (4.1) | 0 (0) | 0.748 |
| Gestational age (weeks), mean (SD) | | 39.0 (1.4) | 39.4 (0.98) | 0.356 |
| Length of stay > 72 h, *n* (%) | | 14 (28.6) | 3 (33.3) | 0.773 |
| Preterm delivery, *n* (%) | | 5 (10.2) | 0 (0) | 0.316 |
| Fetal anomalies present, *n* (%) | | 4 (8.2) | 0 (0) | 0.374 |
| NICU § Admission, *n* (%) | | 3 (6.3) | 0 (0) | 0.441 |

* < 0.05 significant and bolded. † For the nineteen and four in each group who underwent cesarean. ‡ For the thirty and five who underwent cesarean. § NICU = Neonatal intensive care unit.

**Table 6.** Associations between pain and anxiety for participants in the 6-week postpartum survey. Minimum, maximum pain scores in the first and second 24 h time period and for the whole hospitalization, and percentage of time spent in severe pain, per patient survey.

| | Variable | GAD-7 < 10 (*n* = 49) | GAD-7 ≥ 10 (*n* = 9) | *p*-Value * |
|---|---|---|---|---|
| First 24 h post-delivery (*n* = 64) | Maximum pain score ≥ 7, *n* (%) * | 18 (36.7) | 4 (44.4) | 0.661 |
| | % of time spent in severe pain, median (IQR) † | 10 (0–20) | 10 (0–10) | 0.706 |
| | 20% of time or more spent with severe pain, *n* (%) | 13 (26.5) | 1 (11.1) | 0.320 |
| | Lowest pain score ≤ 3, *n* (%) | 43 (87.8) | 6 (66.7) | 0.108 |
| Second 24 h post-delivery | Maximum pain score ≥ 7, *n* (%) | 13 (26.5) | 4 (44.4) | 0.278 |
| | % of time spent in severe pain, median (IQR) | 10 (0–10) | 10 (0–10) | 0.991 |
| | 20% of time or more spent with severe pain, *n* (%) | 10 (20.4) | 1 (11.1) | 0.513 |
| | Lowest pain score ≤ 3, *n* (%) | 45 (91.8) | 8 (88.9) | 0.772 |
| Overall for hospitalization | Maximum pain score ≥ 7, *n* (%) | 23 (46.9) | 4 (44.4) | 0.890 |
| | 20% of time or more spent with severe pain, *n* (%) | 16 (32.6) | 1 (11.1) | 0.192 |
| | Lowest pain score ≤ 3, *n* (%) | 47 (95.9) | 8 (88.9) | 0.381 |
| | Total doses of opioid medications, † | 0 (0–1) | 0 (0–3) | 0.643 |
| | Perceived adequacy of pain control, median (IQR) †‡ | 9 (8–10) | 8 (5–9) | 0.090 |
| | Satisfaction with pain control, median (IQR) †§ | 10 (8–10) | 8 (5–10) | **0.049** |

* Chi-square test used for categorical variables. † Mann–Whitney U-test for ordinal variables ("percentage of time spent in severe pain" was on an ordinal scale) and for non-normally distributed data (example: total opioid doses). ‡ Measured on a 0–10 scale with 0 being completely inadequate and 10 being completely adequate. § Measured on a 0–10 scale with 0 being extremely dissatisfied and 10 being extremely satisfied.

## 4. Discussion

We found that anxiety in the immediate postpartum period was associated with pain in that same timeframe. Those with anxiety determined by the GAD-7 at the time of discharge experienced more severe pain in both the first and second 24 h post-delivery period, and they also spent more of their time in severe pain in both timeframes. Additionally, those

with anxiety had more difficulties doing activities in and out of bed, falling and staying asleep, had less adequate pain control after giving birth, and overall were less satisfied with their pain treatment. By six weeks postpartum, many of these associations were lost in this small study. Our findings support our hypothesis that there is an association between pain and anxiety in the immediate postpartum, although our hypothesis about the relationship between pain and anxiety in the longer-term postpartum period was not supported in this small study.

Our results are similar to one study which demonstrated parturients with increased State-Trait Anxiety Inventory scores reported higher self-rated pain prior to labor analgesia compared to those with low STAI-T scores, although here we were evaluating postpartum pain rather than labor pain [20]. Another study demonstrated a significant correlation between state anxiety and maximum and average labor pain expectancies [21]. Both of these studies suggest a relationship between pain and anxiety in the peripartum period. An additional study demonstrated state anxiety was associated with post-cesarean pain, whereas trait anxiety was not associated with post-cesarean pain [22]. These findings are similar to ours in the sense that anxiety at the time of discharge was associated with increased levels of pain. Here, we did not find an association between preexisting anxiety and increased anxiety or pain in the longer postpartum period. Another study demonstrated a negative perception of a recent birth experience was associated with anxiety symptoms at 2 and 8 months postpartum [23], which is similar to our findings in the immediate postpartum period, but not consistent with the longer-term (6-week) postpartum period results in our small study. Our results differ from a study that demonstrated no significant association between anxiety and the request for epidural anesthesia [24], indicating a lack of association between anxiety and increased pain requiring the request of an epidural, although requesting an epidural is not an adequate surrogate for pain as many factors contribute to the request and desire for epidural analgesia.

Notably, preexisting psychiatric disorders such as anxiety, depression, or both, were not associated with anxiety in the immediate postpartum period in this study, although preexisting depression was associated with anxiety in the longer-term postpartum period (at 6 weeks). Patients with preexisting anxiety did not have higher rates of severe pain, percent of time with severe pain, or less mild pain than those without a history of anxiety, nor did they have different perceptions of inadequate pain control. The lack of association between preexisting anxiety or mood disorders and anxiety or pain in the postpartum period suggests the importance of the relationship between current feelings of anxiety and pain in the immediate postpartum period. The effect of preexisting anxiety on pain in the immediate postpartum period is likely more subtle than was captured in this small sample. Indeed, we previously studied the impact of preexisting anxiety on post-cesarean pain and opioid use and found postpartum people with a preexisting diagnosis of anxiety had higher average pain scores and postpartum opioid use than those without anxiety [25].

Strengths of our study include that parturients were surveyed close to the time of birth in near real-time; thus, the assessment of their self-reported pain had less recall bias. There was a high response rate for the second survey. We also used validated questionaries to assess both anxiety (GAD-7) [16] and pain (APS-POQ) [13,14].

Limitations include our small sample size as there were only 64 parturients that completed the discharge survey. Our original plan was to recruit 400 parturients. This was intended to occur via in-person recruitment with an offer to assist with survey completion by the researcher present on the postpartum unit. However, due to restrictions related to COVID, access to both the electronic health record and postpartum unit was prohibited. Later, access to being physically present on the postpartum unit remained prohibited but access to the health record was permitted; thus, the protocol and regulatory boards were updated to the recruitment approach discussed wherein the rounding provider introduced the study and the researcher called the patient via phone. This heavy modification to the recruitment method contributed to both the small sample size and likely also impacted the diversity of patients recruited. There is a notable potential for bias; patients knew

the survey was about pain and anxiety and this may have influenced who completed the survey or who agreed to be contacted about the study by researchers. Additionally, the response rate for the first survey was low. Generalizability is also limited; the majority of the surveyed patient population was white, married, and had private insurance. This was likely related to the recruitment method which required several steps. Our prior publications indicate that our institution indeed has racial disparities in the management of postpartum pain, and we and others have reported notable differences in known diagnoses of anxiety and depression by race and ethnicity [25–27]. These disparities may also impact who is willing to agree to be contacted by a researcher about a possible research project, and may also impact who is willing to consent to research once they receive an introduction to the study. Recruitment via the original recruitment method, with researchers of multiple racial and ethnic backgrounds approaching patients in-person may have circumvented some of these barriers. Access to the internet is required, but tablet devices are available for use by patients admitted to the postpartum unit which would have allowed for completion of at least the first survey. The ultimately low sample size also limited our ability to perform subgroup analyses such as grouping vaginal deliveries and scheduled cesareans and comparing to those with unplanned cesarean deliveries. However, our prior work demonstrated that unscheduled and emergency cesarean deliveries contribute to anxiety and pain [26]. Our small sample size also limited our ability to control for potentially confounding baseline differences. More people in the group with anxiety had miliary insurance, for example, which may be associated with other baseline differences such as pain related to injuries obtained while in the military and also a myriad of psychological conditions including anxiety [28]. Intraamniotic infection may also impact anxiety as it is an unanticipated new diagnosis that typically occurs intrapartum, which could potentially provoke anxiety although publications supporting this potential could not be identified. Intraamniotic infection could also contribute to fundal tenderness and pain.

Here, we demonstrate that anxiety and pain are associated; this study is insufficient to clarify which is the exposure, and which is the outcome. Anxiety may exacerbate pain or pain may exacerbate anxiety, or there may be a positive feedback loop wherein pain and anxiety amplify each other, or it may be a correlation without causation simply from the overall experience of birthing as it inherently induces stress and pain in the body [3]. Some authors suggest that anxiety may result in catastrophizing pain which subsequently worsens both anxiety and the pain itself [3,29]. Studies that evaluate anxiety during the antepartum period or intrapartum before the onset of significant pain may better elucidate the nature of this relationship. An ideal setting would potentially involve patients undergoing an induction of labor as they would typically not be in pain at the time of admission, but both pain and anxiety could be queried throughout the intrapartum and postpartum course.

The nature of this relationship remains to explored in larger studies, but consideration of the relationship between pain control and anxiety may help optimize patients' postpartum experience. Implementing screening for anxiety during pregnancy provides an opportunity for patients and providers to discuss delivery-related concerns and expectations, how to optimize pain management, and overall to provide targeted counseling for patients [3]. This should be considered for patients as it may be beneficial by tailoring to the needs of the patient. In theory, providing classes, either in-person or online and interactive, which address strategies to manage mental health and wellness or which promote exercise during pregnancy and after could possibly be of benefit to patients, but this has not been borne out as specifically beneficial in the literature [30,31]. There is some evidence that labor and delivery expectations may affect pain perception [3], so discussing these expectations in advance could potentially mitigate the experience of pain. Providing timely, adequate pain management with medications during the intra- and postpartum period may help mitigate the experience of pain. Additionally, the presence of a pregnancy support person or doula has been demonstrated in the literature to reduce anxiety, pain, and the development of post-traumatic stress following childbirth [25,32]. The use of music has

also been implemented to reduce both pain and anxiety levels during labor in nulliparous parturients [3,33], and this could be considered as a strategy postpartum as well. In sum, addressing anxiety may help ameliorate intrapartum and postpartum pain, and this can be addressed with thoughtful discussions with anticipatory guidance, the presence of support persons including doulas, and ancillary measures such as music, yoga, and relaxation techniques [3,25,32,34,35]. Addressing intrapartum and postpartum pain with the use of neuraxial analgesia and regional blocks, such as a transversus abdominis plane block, or other opioid-sparing medications may also reduce anxiety without increasing the use of opioid medications [3].

The relationship between anxiety and pain is established and has real effects on birthing people. Causality is unclear and many have suggested a potential positive feedback loop where anxiety worsens pain and pain then worsens anxiety [3]. Considering this relationship and screening patients for real-time anxiety and pain, and specifically counseling patients about expectations related to pain may be beneficial. Further studies with larger populations are needed to further characterize the relationship between pain and anxiety in this clinical setting.

**Author Contributions:** Conceptualization, J.R.S., Z.N.S., C.G.O., K.M.A.; methodology, J.R.S., Z.N.S., K.M.A.; software J.R.S., K.M.A.; validation: C.G.O., K.M.A.; formal analysis, K.M.A.; investigation, C.G.O., J.R.S., K.M.A.; resources: C.G.O., J.R.S., K.M.A., Z.N.S.; data curation, C.G.O., K.M.A.; writing—original draft preparation, C.G.O., K.M.A.; writing—review and editing, C.G.O., J.R.S., Z.N.S., K.M.A.; visualization, C.G.O., K.M.A.; supervision, K.M.A.; project administration, C.G.O., K.M.A., J.R.S.; funding acquisition, C.G.O., K.M.A. All authors have read and agreed to the published version of the manuscript.

**Funding:** This project was supported by the Herman and Gwendolyn Shapiro Foundation's Summer Research Program at the University of Wisconsin School of Medicine and Public Health and by the Department of Obstetrics and Gynecology at the University of Wisconsin. REDCap access was provided by the Clinical and Translational Science Award (CTSA) program, through the NIH National Center for Advancing Translational Sciences (NCATS), grant UL1TR002373.

**Institutional Review Board Statement:** The study was conducted in accordance with the Declaration of Helsinki, and approved by the Institutional Review Board (or Ethics Committee) of UnityPoint Health-Meriter (IRB# 2019-016, 18 September 2019).

**Informed Consent Statement:** Informed consent was obtained from all subjects involved in the study.

**Data Availability Statement:** The data presented in this study are available on request from the corresponding author.

**Acknowledgments:** We would like to acknowledge the OBGYN residents, CNMs, and NPs at UnityPoint Meriter Hospital for their support and assistance with patient recruitment for this study. We thank the University of Wisconsin School of Medicine and Public Health Herman and Gwendolyn Shapiro Foundation and the University of Wisconsin Department of Obstetrics and Gynecology for supporting this project. We thank Robert Koehler for his assistance with the literature review, Carla Griffin for her support of this project, and all the patients and families who participated in this project.

**Conflicts of Interest:** Zachary N. Stowe has received salary and research support from the NIH and the CDC. In the past 36 months, ZNS served on the advisory board for Sage Therapeutics, and prior to 2016, he received clinical trial support from Janssen Pharmaceuticals and Sage Therapeutics. Prior to 2009, ZNS received research support and consultation honorarium from GlaxoSmithKline, Pfizer, and Wyeth Corporations and received speaker honoraria from these companies and Eli Lilly and Forest Corporations. The funders had no role in study design, data collection and analysis, decision to publish, or preparation of the manuscript.

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
