# Peer review of "Assessment of In-Hospital Pain Control after Childbirth and Its Correlation with Anxiety in the Postpartum Period: A Cross-Sectional Study at a Single Center in the USA"

_2673-3897, doi:10.3390/reprodmed3040026_

Round 1

Reviewer 1 Report

The article aims to examine the relationship between postpartum pain control and postpartum anxiety. The findings are clinically and scientifically relevant. However, I suggest the following minimal changes to improve the manuscript:

·        Statistical analysis: Consider including a post hoc power analysis to support your data.

·        Internal consistency of the questionnaires should be reported by calculating traditional coefficients such as Cronbach's alpha and/or McDonald's omega.

Nonetheless, it is a potentially important contribution to the literature. Further, I believe that this article will be useful for the readers of Reproductive Medicine.

Author Response

Reviewer 1

·        Statistical analysis: Consider including a post hoc power analysis to support your data.

Thank you.

We have added a post hoc power analysis using the results we obtained. The prevalence of severe pain was 3.7 times higher than we expected, and differences between groups were higher than we expected, thus the requisite sample size was smaller than we had estimated at the time that the study was designed.

Lines 142-143

·        Internal consistency of the questionnaires should be reported by calculating traditional coefficients such as Cronbach's alpha and/or McDonald's omega.

Thank you.

We have clarified that these are validated questionnaires that have been used in other studies and we have added citations indicating their internal consistency, reliability, and validity.

Lines 93 and 101-102

Reviewer 2 Report

Thank you for giving me the opportunities to review this manuscript.

This article described the association between pain and anxiety in the postpartum period.

I think it is necessary to revise the manuscript.

1) Please change the title as "Assessment of in-hospital pain control after childbirth and its correlation with anxiety in the postpartum period; a cross-sectional study at a single center in the U.S.".

2) I think it is a cross-sectional study. First, the authors tried to assess the correlation of anxiety with pain control. Second, they assessed the association only at the one time point. Third, the time period seemed to be very short. Fourth, the authors actually described that "this study is insufficient 306 to clarify which is the exposure, and which is the outcome." (that is because this was not a cohort study to assess causality). If so, please add it in the title and the abstract. In addition, please change "This survey-based prospective cohort study" as "This survey-based cross-sectional study" in the method section. If this was a cohort study, please describe all the time point and the PECO (the population,  the exposure, the control, and the outcome) clearly.

3) If any, please describe all predictors, potential confounders, and effect modifiers.

4) Please describe all statistical methods used to control for confounding.

5) Please describe how missing data was handled for statistical analysis (the LOCF or the multiple imputation?). 

6) Please indicate number of participants with missing data for each variables of interest. If this was a cohort study, please summarize follow-up time (eg, average and total amount).

7) If possible,  please describe unadjusted estimates and, if applicable, confounder-adjusted estimates and their precision (eg, 95% confidential interval). Please make clear which confounders were adjusted for and why they were included. For example, insurance provider might be a strong confounder. I think that the type of insurance may affect anxiety related to economy. How to control pain medically might be expensive in some cases.  Furthermore,  intraamniotic infection may affect both anxiety and pain. Please explain these aspects in the result and discussion sections.

8) It is better to add the table to explain there were no differences in demographics between those with GAD scores >10 and those <10 at 6 weeks postpartum.

I think it is necessary to revise the manuscript.

Author Response

Responses to Reviewer 2

Reviewer 2

 1) Please change the title as "Assessment of in-hospital pain control after childbirth and its correlation with anxiety in the postpartum period; a cross-sectional study at a single center in the U.S.".

Thank you. We have made this change.

Title

2) I think it is a cross-sectional study. First, the authors tried to assess the correlation of anxiety with pain control. Second, they assessed the association only at the one time point. Third, the time period seemed to be very short. Fourth, the authors actually described that "this study is insufficient 306 to clarify which is the exposure, and which is the outcome." (that is because this was not a cohort study to assess causality). If so, please add it in the title and the abstract. In addition, please change "This survey-based prospective cohort study" as "This survey-based cross-sectional study" in the method section. If this was a cohort study, please describe all the time point and the PECO (the population,  the exposure, the control, and the outcome) clearly.

Thank you. We have updated this throughout the manuscript.

Title, Abstract, Lines 17-18, 59

3) If any, please describe all predictors, potential confounders, and effect modifiers.

Thank you.

The only differences in baseline characteristics were differences in payor and intraamniotic infection. Unfortunately, due to our low sample size we were not able to control for this in our outcomes analysis.

4) Please describe all statistical methods used to control for confounding.

Thank you.

Due to our low sample size, these were not performed.

5) Please describe how missing data was handled for statistical analysis (the LOCF or the multiple imputation?). 

Thank you.

Only the surveys that were completed were included. Coincidentally, all surveys that were started were completed so there were no partly completed surveys or partial datasets. This was the case for both the discharge surveys and the 6-week postpartum follow-up surveys.

However, for those who completed a discharge survey but not a 6-week postpartum survey, only the 6-week postpartum surveys that were started/ completed were included in that analysis (tables 5-6).

6) Please indicate number of participants with missing data for each variables of interest. If this was a cohort study, please summarize follow-up time (eg, average and total amount).

There were 6 weeks of follow-up time with 58 people completing the six-week postpartum survey. No one was followed for longer than 6 weeks. We have added the average time after discharge from the hospital that participants completed the 6-week survey and also the range.

Lines 228-229

7) If possible,  please describe unadjusted estimates and, if applicable, confounder-adjusted estimates and their precision (eg, 95% confidential interval). Please make clear which confounders were adjusted for and why they were included. For example, insurance provider might be a strong confounder. I think that the type of insurance may affect anxiety related to economy. How to control pain medically might be expensive in some cases.  Furthermore,  intraamniotic infection may affect both anxiety and pain. Please explain these aspects in the result and discussion sections.

Thank you. Due to the low sample size, no adjustment for confounding variables was performed.

The confounding effect of payor has been added as a limitation as has intraamniotic infection.

Lines 309-316

8) It is better to add the table to explain there were no differences in demographics between those with GAD scores >10 and those <10 at 6 weeks postpartum.

Thank you.

These tables have been added.
Tables 5-6

Round 2

Reviewer 2 Report

Thank you for revising the manuscript.

I think this manuscript would be suitable for publication in this journal.